# Comparison of Genotoxicity and Pulmonary Toxicity Study of Modified SiO$_2$ Nanomaterials

Yng-Tay Chen [1,*] , Po-Yi Lue [2] , Po-Wei Chen [2], Pin-Ju Chueh [3,*] , Fuu-Jen Tsai [4,5,6] and Jiunn-Wang Liao [2,*]

[1] Graduate Institute of Food Safety, National Chung Hsing University, Taichung 40227, Taiwan
[2] Graduate Institute of Veterinary Pathobiology, National Chung Hsing University, Taichung 40227, Taiwan; k810430k@hotmail.com (P.-Y.L.); jsps050208@gmail.com (P.-W.C.)
[3] Graduate Institute of Biomedical Science, College of Life Science, National Chung Hsing University, Taichung 40227, Taiwan
[4] Human Genetic Center, Department of Medical Research, China Medical University Hospital, Taichung 406040, Taiwan; d0704@mail.cmuh.org.tw
[5] School of China Medicine, China Medical University, Taichung 406040, Taiwan
[6] Department of Biotechnology and Bioinformatics, Asia University, Taichung 41354, Taiwan
* Correspondence: ytchen101@dragon.nchu.edu.tw (Y.-T.C.); pjchueh@dragon.nchu.edu.tw (P.-J.C.); jwliao@dragon.nchu.edu.tw (J.-W.L.)

**Abstract:** Surface-modified nano-SiO$_2$ is a common additive in many products. However, the safety of nano-SiO$_2$ products under various modifications is still unclear. In this study, we investigated the genotoxicity and acute pulmonary toxicity of nano-SiO$_2$ with or without modification. The samples used in this study included: sample A (SA, 55.16 nm, 411.3 mg/mL), modified sample A (mSA, 82.29 nm, 37.7 mg/mL), sample B (SB, 22 nm, 358.0 mg/mL), and modified sample B (mSB, 86.64 nm, 37.7 mg/mL). In the genotoxicity study, we conducted an Ames test, chromosomal aberration test (CA), and a micronucleus (MN) test. The SA, mSA, and mSB groups showed negative results in all these genotoxicity tests. Only SB showed a weakly positive reaction in these assays, but the genotoxicity could be reversed after S9 metabolism or modification. In the acute pulmonary toxicity test, the rats were given an intratracheal instillation (IT) (0.5 mL/kg) of diluted samples and sacrificed after 1 or 14 days. The mortality rate, number of leukocytes and cytokines of TNF-$\alpha$ in the bronchoalveolar lavage fluid (BALF), and the pathology in the lungs were determined. The results revealed that mSA posed acute toxicity in rats. After modification, the pulmonary toxicity was increased in mSA but decreased in mSB on Day 1, and no significant difference was observed on Day 14. In conclusion, there was no observed genotoxicity in either SA or SB, while mSA posed acute inhalation toxicity to rats that decreased in mSB after modification. This indicates that the decrease in pH level in SA and decrease in the solid content in SB are considered after the trifluorosilane surface-modified amorphous nano-silica.

**Keywords:** nanoparticle modification; genotoxicity; pulmonary toxicity; silica



## 1. Introduction

The definition of a nanomaterial is an object with at least one dimension measuring 1–100 nanometers [1]. Recently, technological developments have resulted in the applications of nanomaterials becoming more and more diverse. Among those nanomaterials, surface-modified nano-SiO$_2$ is a common additive in many products, such as printer toners, paint coating, animal feed, pesticides, cosmetics, target drugs, and medicine additives, etc. [2]. However, with increased consumption of nano-SiO$_2$, there are increased chances of people coming into contact with these nanoparticles. Inhalation, cutaneous absorption, oral ingestion, and intravenous injection are common exposure routes of nanomaterials [3]. Of all these exposure routes, inhalation is the most common one [4]. However, the safety of exposure to nanoparticles is not fully understood. Nanomaterials may spread in the air and be inhaled into the lungs [5]. When the dosage of particles exceeds the particle

clearance ability of the lungs, the term "particle overload" is used to characterize these conditions. The hallmark of the particle-overloaded lung is an impairment of alveolar macrophage (AM)-mediated lung clearance which has been demonstrated in all species tested so far, and eventually leads to the accumulation of excessive lung burdens [6]. When particle overload occurs, it may induce many biological hazards that have been mentioned in previous studies, such as carcinogenesis [7], genotoxicity [8], and pulmonary toxicity [9].

The toxicity of nano-silica in the lungs may vary according to different particle sizes and crystalline form [10]. Nano-silicas with smaller sizes induce more reactive oxygen species (ROS) in the lung [11,12]. The ROS may interfere with cell function by interacting with macrophages and alveolar cells and induce DNA or protein damage [13,14]. Therefore, nanomaterials may induce genotoxicity through direct or indirect pathways [15–21]. In recent research, surface modification of nano-silica has been developed. Surface modification may improve the dispersion [22], surface activity, and compatibility with other materials of nano-powder. Therefore, modification may produce new physical, chemical, and mechanical properties and functions in the applications of nanomaterials. In previous research, when compared to unmodified nonporous silica nanoparticles, amine modification of nonporous silica nanoparticles reduced murine lung inflammation and improved the overall biocompatibility of the nanomaterial [23].

In this study, we focused on the evaluation of the toxicity and biocompatibility of trifluorosilane surface-modified amorphous nano-silica. To evaluate the changes of toxicity before and after the nano-silica modification, we conducted genotoxicity and rat pulmonary toxicity tests.

## 2. Materials and Methods

### 2.1. Samples

The test articles used in this study are four kinds of surface-modified and unmodified nano-silicas, namely, sample A (SA), modified sample A (mSA), sample B (SB), and modified sample B (mSB). The silica nanoparticles were obtained by hydrolysis of tetraethoxysilane (TEOS) in an ethanol medium. The nanomaterials were obtained from the Industrial Technology Research Institute (ITRI, Zhudong, Taiwan). R.O.C. ITRI also provided us with the physical and chemical characteristic information of those materials. The structure of test particles was examined using transmission electronic microscopy (TEM). After diluting in purified water, samples were placed over a copper grid coated with carbon film then stained with 2% phosphotungstic acid. The samples were air-dried prior to placement in the TEM instrument for analysis. The shapes and sizes of nano-silicas were determined using a TEM, which showed that particles diameters in four different preparations were in the nanosized range (1–100 nm). The average particle sizes of SA, mSA, SB, mSB are 55.16, 82.29, 22, and 86.64 nm, and the solid content are 41.13, 3.77, 35.8, 3.77%, respectively (Figure 1; Table 1). The nanomaterials were prepared in their powder form, which was stable and kept away from light at all times. However, to reduce instability that may occur during our experiments, we only weighed out enough powder to make the solution for one experiment and used them immediately.

**Table 1.** The chemical properties of unmodified and modified nano-silica.

| Sample | Principal Component | Solid Content (%) | Diameter (nm) | pH |
|---|---|---|---|---|
| Sample A | Colloidal $SiO_2$ | 41.13 | 55.16 | 7~9 |
| Modified sample A | Sample A TEOS [a] $H_3FSi$ | 3.77 | 82.29 | 2~4 |
| Sample B | Colloidal $SiO_2$ | 35.8 | 22 | 7~9 |
| Modified sample B | Sample B TEOS [a] $H_3FSi$ | 3.77 | 86.64 | 2~4 |

[a] TOES: tetraethoxysilane.

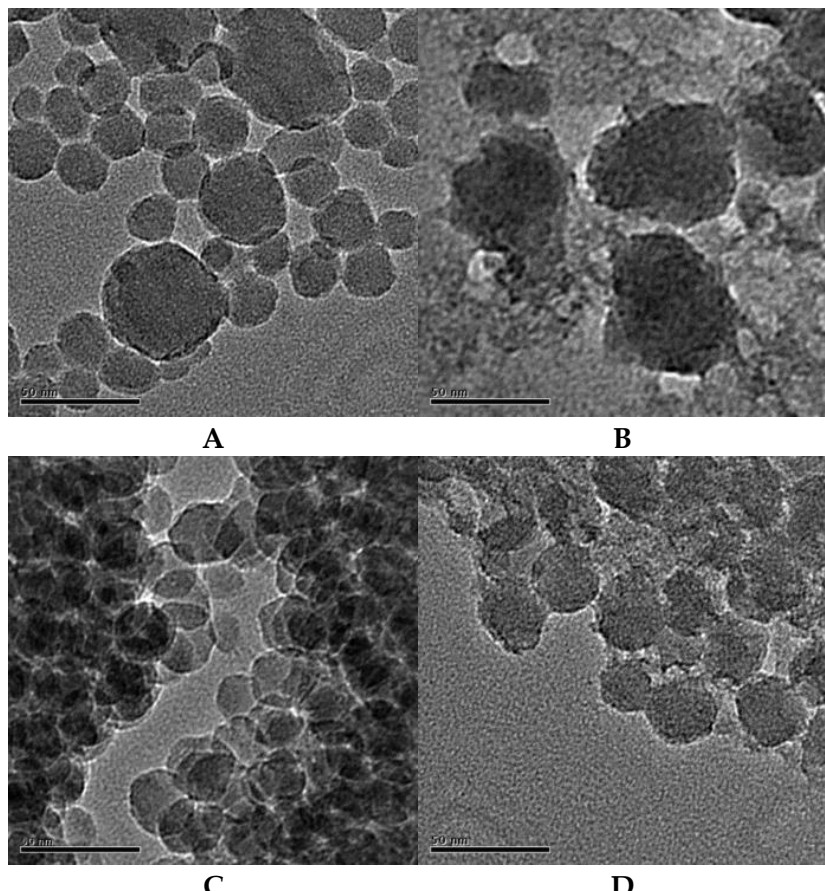

**Figure 1.** The TEM images of the test articles: sample A (**A**), modified sample A (**B**), sample B (**C**), modified sample B (**D**).

### 2.2. Animals

Adult male SPF Sprague Dawley (SD) rats, 4 weeks of age, weighing 150–175 g (for the peripheral blood micronucleus test, MN test), and 8 weeks of age, weighing 275–300 g (for the acute pulmonary toxicity test) at the start, were purchased from BioLASCO Taiwan, co., Ltd. (Taipei, Taiwan). The animals were housed in standard suspension stainless steel cages under specific environmental conditions (22 ± 1 °C, 30–70% relative humidity, 12 h light/dark cycle). Autoclaved Rat Chow (Purina 5010, St. Louis MO, USA) and reverse-osmosis water were available ad libitum. This study was approved by the Institutional Animal Care and Use Committee (IACUC) of National Chung Hsing University (IACUC: 104-136).

### 2.3. In Vitro Genotoxicity Study

#### 2.3.1. Ames Test

Briefly, the Ames test is based upon *Salmonella typhimurium* TA98 and TA100 strains that have mutations on the HIS operon. The mutation makes the TA strains unable to generate the histidine that is essential for bacterial replication [24]. TA98 and TA100 strains were used to detect the mutagenicity of SA, mSA, SB, and mSB in the presence or absence of rat liver S9 fraction. The maximum bactericidal dose of test article was determined by the bactericidal test (data not shown), and the four doses obtained by twofold continuous dilution were added with the mixture of 0.2 mL of 0.5 mM histidine (Merck, kGaA, Darmstadt, Germany) and 0.5 mM biotin and 100 μL TA strains. In the assay with S9 bioactivation, 0.2 mL of S9 mixture were added. Then, the solutions were added into 45 °C 2 mL 0.75% soft agar containing 0.5% NaCl, vortexed well, and poured into minimal glucose agar plates (MA) comprised of agar, glucose solution, magnesium sulfate,

citric acid monohydrate, potassium phosphate dibasic anhydrous, and sodium ammonium phosphate. The plate was incubated at 37 °C overnight. The number of colonies per plate was counted, and the ratio of the number of histidine-revertant colonies to the number of spontaneous revertant colonies for the negative controls was obtained.

The results of the Ames test were analyzed by Student's *t*-test and Microsoft Excel to compare the data between groups. A significant difference was determined if there was a more than twofold difference in the numbers between the negative control and treated groups at $p < 0.05$.

### 2.3.2. Chromosomal Aberration (CA) Test

The CA test is an in vitro genotoxicity test that is conducted using a mammalian cell line [25]. Briefly, the Chinese hamster ovary K1 (CHO-K1) cell line (BCRC 60006, Culture Collection and Research Center, Food Industry Research and Development Institution, Hsinchu, Taiwan) was grown in Ham's F-12 medium supplemented with 10% fetal bovine serum, penicillin (100 unit/mL), and streptomycin (100 µg/mL) at 37 °C in a 5% $CO_2$ humidified incubator. CHO-K1 cells were seeded in 25 cm² flasks at a density of 2.5 × 105 cells/mL culture medium and incubated for 24 h. Then, 30% lethal concentration ($LC_{30}$) doses of the test articles, which were determined by the cytotoxicity test (data not shown), were added and incubated for 24 h with or without S9. During the last 6 h of culturing, cell cycles were stopped after exposure with 10 µL colcemid (10 µg/mL) (Sigma, Darmstadt CAS No. 64-86-8). The CHO-K1 cells were detached with trypsin (0.05%)-EDTA (0.53 mM) (Gibco, Waltham, MA, USA) and fixed by fixation solution (methanol: acetate = 3:1, ALPS, CSC, Taipei Taiwan) after being swollen by 0.54% KCl (Merck, Darmstadt, Germany). The fixed cells were dropped on slides, stained by Diff-Quik stain (Sysmex, Hyogo, Japan) and observed under light microscopy. A total number of 100 metaphases of cells were counted under oil-immersed microscopy for each dosage. The number of cells with damaged chromosomes was recorded, from which the rate of mutation was calculated. Aberration rate (%) = (number of cells with damage chromosomes/100) × 100.

The results of the CA test studies were analyzed by Student's *t*-test and Microsoft Excel to compare the data between groups. A significant difference in the aberration rate was determined if the difference between the negative control and treated groups was more than twofold at $p < 0.05$.

### 2.4. In Vivo Genotoxicity and Acute Pulmonary Toxicity Test

#### 2.4.1. Acute Pulmonary Toxicity Test (1 D)

Thirty-two male SD rats were randomly divided into six groups, as shown in Table 2. High mortality within 24 h after IT was noticed in the mSAH group animals in the acute pulmonary toxicity test (1 D). Therefore, in the rat peripheral blood micronucleus test and acute pulmonary toxicity test (14 D), to ensure all the animals would survive to the end of the study, the dose level of mSA group was reduced to 320×. Then, to be comparable with the mSA group, the dose level of SA group was also reduced to 320×. Because of the lethal effects of mSA, two subgroups of low (mSAL) and high doses (mSAH) were designated. The rats were given a single intratracheal instillation and sacrificed at 1 day after the administration. The lungs were excised for the bronchoalveolar lavage fluid (BALF) analysis and histopathological examination. To obtain the BALF, after the lungs were weighed, the right lung was subjected to lavage five times with 6 mL 4 °C phosphate-buffered saline (PBS). Then, the BALF was centrifuged (2 min at 400× $g$) and the acellular supernatant was analyzed for total protein and tumor necrosis factor-$\alpha$ (TNF-$\alpha$). The total number of BALF white blood cells (WBC) were counted by a hematology analyzer (Toa Medical Electronics Co., Sysmex K-4500, Hyogo, Japan) with trypan blue stain, separated by cytocentrifuge (GMI, Thermo Shandon Cytospin 3, New York, NY, USA), placed on glass microscope slides, and then stained by Diff-Quik stain. The ratio of neutrophils, eosinophils, lymphocytes, and macrophages in the BALF WBCs was obtained from 100 WBCs per slide by microscopic observation. For histopathological evaluation, after ligation

of the trachea and the right bronchus, the left lung was instilled with 1 mL 10% neutral buffered formalin (NBF), and the entire lobe was immersed in 10% NBF for at least 1 day. The fixed lung lobes were embedded in paraffin, and 5-μm-thick sections were cut from the lobe by microtome (Leica RM2145, Nussloch, Germany) then stained with hematoxylin and eosin. The histopathological score evaluation was performed as previously described [26].

**Table 2.** Study design of nano-silica in the acute pulmonary toxicity.

| Group Code | Substance | Dose | | Animal Number |
| --- | --- | --- | --- | --- |
| | | (mg/kg) | (mL/kg) | |
| NC [1] | R.D.W. | - | 0.5, i.t. | 6 |
| SA | 80× diluted SA | 2.58 | 0.5, i.t. | 6 |
| mSAH | 80× diluted mSA | 0.24 | 0.5, i.t. | 5 |
| mSAL | 320× diluted mSA | 0.06 | 0.5, i.t. | 3 |
| SB | 80× diluted SB | 2.24 | 0.5, i.t. | 6 |
| mSB | 80× diluted mSB | 0.24 | 0.5, i.t. | 6 |

[1] NC: negative control, R.D.W.; SA: sample A; mSAH: modified sample A high dose; mSAL: modified sample A low dose; SB: sample B; mSB: modified sample B; i.t.: intratracheal instillation.

### 2.4.2. Rat Peripheral Blood Micronucleus Test and Acute Pulmonary Toxicity Test (14 D)

Twenty-nine male rats were randomly divided into six groups, as shown in Table 3. At 48 and 72 h after the test, articles were administered to the rats and fresh peripheral blood was obtained by orbital sinus sampling under 2% isoflurane anesthesia. The blood samples were prepared by using the rat MicroFlow (PLUS) Kit purchased from Litron, lab., Rochester, NY, USA. The blood was analyzed by flow cytometry following the manufacturer's guidelines for the MicroFlow (PLUS) Kit to determine the rate between normochromatic erythrocytes (NCE) and reticulocytes (RET) and the frequency of micronucleated reticulocytes (Mn-RET).

**Table 3.** Treatment groups and doses of the rat peripheral blood micronucleus test and acute pulmonary toxicity on Day 14.

| Group Code | Substance | Dose | | Animal Number |
| --- | --- | --- | --- | --- |
| | | (mg/kg) | (mL/kg) | |
| NC [1] | R.D.W. | - | 0.5, i.t. | 5 |
| PC | Cyclophosphamide | 20 | 10, i.p. | 4 |
| SA | 320× diluted SA | 0.645 | 0.5, i.t. | 5 |
| mSA | 320× diluted mSA | 0.06 | 0.5, i.t. | 5 |
| SB | 80× diluted SB | 2.24 | 0.5, i.t. | 5 |
| mSB | 80× diluted mSB | 0.24 | 0.5, i.t. | 5 |

[1] NC: negative control, R.D.W.; PC: positive control, cyclophosphamide; SA: sample A; mSA: modified sample A; SB: sample B; mSB: modified sample B; i.t.: intratracheal instillation; i.p.: intraperitoneal injection.

The results of the MN test were analyzed by Student's *t*-test and Microsoft Excel to compare the data between groups. A significant difference in the frequency of Mn-RET was determined if the difference between the negative control and treated groups was more than twofold at $p < 0.05$. At 14 days after intratracheal instillation, the rats were sacrificed and the brain, heart, liver, spleen, lungs, kidneys, thymus, testis, and adrenal glands were excised, weighed, and fixed with 10% NBF. The BALF was obtained and analyzed by the same method as previously mentioned.

### 2.5. Statistical Analysis

All the results in the acute pulmonary toxicity test were analyzed by Student's *t*-test and Microsoft Excel to compare the data between groups. A significant difference in all the parameters at $p < 0.05$ was compared between the treatment and negative control groups.

## 3. Results

### 3.1. Ames Test

In the Ames test, five different concentrations of SA, mSA, SB, and mSB were tested on TA98 and TA100 strains with and without S9 metabolic activation. For the TA98 strain, no tested samples increased the number of revertant colonies with or without S9 fraction. For the TA100 strain, SA and mSB did not increase the number of revertant colonies with or without S9 fraction. In the 0.4713 mg/plate concentration of mSA without S9 metabolism, the number of revertant colonies was significantly increased with 1.05 times the negative control. The number of revertant colonies in the mSA group was not more than twofold of the negative control, indicating a negative result in the Ames test. In the 0.4713, 0.9425, and 1.885 mg/plate concentrations of SB without S9 metabolism, the number of revertant colonies was significantly increased, with 1.57, 1.75, and 2.47 times that of the negative control, respectively. Due to the number of revertant colonies in the SB group increasing in a dose-dependent manner to over twofold that of the negative control at the 1.885 mg/plate concentration, SB induced a positive result in the Ames test with S9 metabolism. However, the number of revertant colonies of the TA100 strain did not significantly increase in the mSA and SB group with S9 metabolism (Tables 4 and 5). Overall, the data of the Ames tests showed that SB might induce genotoxicity in the TA100 strain. However, genotoxicity might be reduced with S9 metabolism.

**Table 4.** Revertant changes of SA, mSA, SB, and mSB in *Salmonella* TA98 mutagenicity test.

| Sample | | SA | mSA | SB | mSB |
|---|---|---|---|---|---|
| | | **Number of Revertants (Colony, CFU/Plate)** | | | |
| | NC [1] | 22.7 ± 2.1 [2] | 28.0 ± 5.1 | 25.0 ± 3.7 | 25.0 ± 3.7 |
| | 4NQO (positive control) | 213.7 ± 5.7 * | 586.3 ± 75.9 * | 468.3 ± 7.0 * | 468.3 ± 7.0 * |
| TA98/without S9 activation (mg/plate) | 0.0295 | - | 31.3 ± 0.9 | - | 20.7 ± 1.2 |
| | 0.0589 | 22.0 ± 1.4 | 26.3 ± 5.4 | 25.0 ± 6.4 | 22.7 ± 3.1 |
| | 0.1178 | 20.3 ± 0.5 | 23.7 ± 1.7 | 23.0 ± 2.9 | 21.7 ± 2.5 |
| | 0.2356 | 23.0 ± 1.4 | 27.3 ± 1.2 | 21.0 ± 0.8 | 21.0 ± 2.2 |
| | 0.4713 | 23.7 ± 2.6 | 37.7 ± 3.7 | 25.3 ± 0.9 | 22.7 ± 1.2 |
| | 0.9425 | 22.7 ± 1.9 | - | 20.3 ± 0.5 | - |
| Sample | | SA | mSA | SB | mSB |
| | | **Number of Revertants (Colony, CFU/Plate)** | | | |
| | NC | 27.3 ± 2.1 | 27.3 ± 2.1 | 34.0 ± 3.7 | 34.0 ± 3.7 |
| | 2-AA (positive control) | 2590.7 ± 138.9 * | 2590.7 ± 138.9 * | 1829.3 ± 236.1 * | 1829.3 ± 236.1 * |
| TA98/with S9 activation (mg/plate) | 0.0295 | - | 26.7 ± 2.5 | - | 34.7 ± 4.6 |
| | 0.0589 | 28.7 ± 1.7 | 26.3 ± 3.1 | 33.3 ± 2.6 | 35.7 ± 0.9 |
| | 0.1178 | 31.3 ± 2.5 | 26.7 ± 3.4 | 34.0 ± 3.7 | 29.3 ± 1.2 |
| | 0.2356 | 27.7 ± 2.9 | 28.7 ± 3.4 | 35.0 ± 2.2 | 32.0 ± 2.2 |
| | 0.4713 | 26.0 ± 0.8 | 26.0 ± 3.7 | 31.7 ± 2.6 | 31.3 ± 4.2 |
| | 0.9425 | 27.7 ± 2.1 | - | 29.3 ± 2.6 | - |

[1] NC: negative control, R.D.W.; 4NQO: 4-nitroquinoline 1-oxide, positive control; SA: sample A; mSA: modified sample A; SB: sample B; mSB: modified sample B; 2-AA: anthranilic acid, positive control. [2] Data are expressed as the mean ± SD (n = 3). * Significantly different between blank and treatment ($p < 0.05$).

**Table 5.** Revertant changes of SA, mSA, SB, and mSB in *Salmonella* TA100 mutagenicity test with or without S9.

| Sample | | SA | mSA | SB | mSB |
|---|---|---|---|---|---|
| | | Number of Revertants (Colony, CFU/Plate) | | | |
| NC [1] | | $194.7 \pm 2.1$ [2] | $194.7 \pm 2.1$ | $143.0 \pm 6.7$ | $187.7 \pm 7.8$ |
| Sodium azide (positive control) | | $1090.3 \pm 52.1$ * | $1090.3 \pm 52.1$ * | $1819.0 \pm 196.5$ * | $1665.3 \pm 32.3$ * |
| TA100/without S9 activation (mg/plate) | 0.0295 | $195.0 \pm 10.7$ | $188.3 \pm 10.7$ | - | - |
| | 0.0589 | $188.0 \pm 11.6$ | $201.7 \pm 15.5$ | - | $182.3 \pm 6.2$ |
| | 0.1178 | $208.0 \pm 11.4$ | $195.7 \pm 4.7$ | $153.0 \pm 7.8$ | $192.3 \pm 9.2$ |
| | 0.2356 | $199.3 \pm 7.0$ | $184.7 \pm 6.2$ | $162.7 \pm 10.6$ | $189.0 \pm 4.5$ |
| | 0.4713 | $214.0 \pm 13.4$ | $204.7 \pm 3.7$ * | $225.0 \pm 3.6$ * | $172.7 \pm 8.3$ |
| | 0.9425 | - | - | $251.3 \pm 13.5$ * | $172.3 \pm 2.1$ |
| | 1.885 | - | - | $354.0 \pm 5.9$ * | - |
| Sample | | SA | mSA | SB | mSB |
| | | Number of Revertants (Colony, CFU/Plate) | | | |
| NC | | $145.0 \pm 10.6$ | $145.0 \pm 10.6$ | $185.0 \pm 8.3$ | $185.0 \pm 8.3$ |
| 2-AA (positive control) | | $2449.3 \pm 216.6$ * | $2449.3 \pm 216.6$ * | $3288.3 \pm 181.5$ * | $3288.3 \pm 181.5$ * |
| TA100/with S9 activation (mg/plate) | 0.0295 | $135.3 \pm 9.4$ | $128.3 \pm 6.9$ | - | - |
| | 0.0589 | $134.7 \pm 7.7$ | $130.0 \pm 6.4$ | - | $192.7 \pm 9.3$ |
| | 0.1178 | $128.7 \pm 2.1$ | $137.7 \pm 10.3$ | $168.7 \pm 9.0$ | $175.0 \pm 5.7$ |
| | 0.2356 | $136.7 \pm 8.7$ | $171.7 \pm 9.5$ | $192.3 \pm 9.0$ | $178.0 \pm 7.5$ |
| | 0.4713 | $129.0 \pm 5.1$ | $150.3 \pm 8.7$ | $182.3 \pm 6.9$ | $177.7 \pm 5.2$ |
| | 0.9425 | - | - | $193.0 \pm 4.2$ | $180.7 \pm 7.1$ |
| | 1.885 | - | - | $205.3 \pm 9.0$ | - |

[1] NC: negative control, R.D.W.; 4NQO: 4-nitroquinoline 1-oxide, positive control; SA: sample A; mSA: modified sample A; SB: sample B; mSB: modified sample B; 2-AA: anthranilic acid, positive control. [2] Data are expressed as the mean $\pm$ SD (n = 3). * Significantly different between blank and treatment ($p < 0.05$).

*3.2. CA Test*

The LC 30 values of SA, mSA, SB, and mSB were 1.387, 0.778, 0.353, and 1.569 mg/mL, respectively. These figures were based on cytotoxicity tests (data not shown). First, all the test articles were tested at concentrations of LC 30 in the absence of a rat S9 metabolic activation system. The results revealed that there was no significant difference in the frequencies of chromosomal aberrations in the SA, mSA, or mSB groups compared to the negative control. However, the SB-treated CHO-K1 cells showed significantly increased frequencies of chromosomal aberrations when compared to the control group. Because the frequencies of chromosomal aberrations in the SB group were increased by about 3.7-fold from that of the negative control, the results of the CA test in the SB group with rat S9 metabolism were positive (Table 6). Second, in the presence of a rat S9 metabolic activation system, the results revealed no significant difference in the frequencies of chromosomal aberrations in the treatment groups compared to the negative control (Table 7). Overall, in the CA test, SA, mSA, and mSB revealed negative results with or without rat S9 metabolism. However, SB without rat S9 metabolism showed genotoxicity to CHO-K1 cells, indicating the genotoxicity of SB to CHO-K1 cells might be reduced with rat S9 metabolism.

**Table 6.** Frequency of chromosomal aberration without S9 metabolism of unmodified and modified nano-silica in CHO-K1 cells.

| Group | Dosage (mg/mL) | Frequency of Aberration (%) | Chromatic Type | | | Chromosome Type | |
|---|---|---|---|---|---|---|---|
| | | | Deletion | Intrachange | Interchange | Ring | Gap |
| NC [1] | 0 | 2.7 ± 1.5 | 2.7 ± 1.5 | 0.0 ± 0.0 | 0.0 ± 0.0 | 0.0 ± 0.0 | 0.0 ± 0.0 |
| Mitomycin C | 2.5 | 18.0 ± 2.6 * | 13.3 ± 6.1 * | 0.0 ± 0.0 | 3.3 ± 0.6 | 0.7 ± 0.6 | 5.0 ± 3.5 * |
| SA | 1.387 | 2.3 ± 1.5 | 1.7 ± 0.6 | 0.0 ± 0.0 | 0.0 ± 0.0 | 0.0 ± 0.0 | 0.7 ± 1.2 |
| mSA | 0.778 | 3.0 ± 2.0 | 2.7 ± 2.1 | 0.0 ± 0.0 | 0.0 ± 0.0 | 0.0 ± 0.0 | 0.3 ± 0.6 |
| SB | 0.353 | 9.7 ± 1.5 *,a | 6.0 ± 2.0 [a] | 0.0 ± 0.0 | 0.7 ± 1.2 | 0.0 ± 0.0 | 4.0 ± 1.0 *,a |
| mSB | 1.569 | 2.7 ± 1.5 [a] | 2.0 ± 1.0 [a] | 0.0 ± 0.0 | 0.0 ± 0.0 | 0.0 ± 0.0 | 0.7 ± 1.2 [a] |

[1] NC: negative control, R.D.W. 10% *v/v*; SA: sample A, 10% *v/v*; mSA: modified sample A, 10% *v/v*; SB, sample B, 10% *v/v*; mSB, modified sample B, 10% *v/v*. * Significant difference in compared with the negative control and treated groups at *p* < 0.05. [a] Significant difference in comparison with the SB and mSB groups at *p* < 0.05.

**Table 7.** Frequency of chromosomal aberration with S9 metabolism of unmodified and modified nano-silica in CHO-K1 cells.

| Group | Dosage (mg/mL) | Frequency of Aberration (%) | Chromatic Type | | | Chromosome Type | |
|---|---|---|---|---|---|---|---|
| | | | Deletion | Intrachange | Interchange | Ring | Gap |
| NC [1] | 0 | 2.7 ± 2.1 | 1.3 ± 1.2 | 0.0 ± 0.0 | 0.3 ± 0.6 | 0.0 ± 0.0 | 1.0 ± 1.0 |
| Cyclophophamide | 25 | 12.7 ± 3.1 * | 13.7 ± 4.7 * | 0.0 ± 0.0 | 0.3 ± 0.6 | 0.0 ± 0.0 | 1.7 ± 0.6 |
| SA | 1.387 | 3.3 ± 0.6 | 2.0 ± 0.0 | 0.0 ± 0.0 | 0.3 ± 0.6 | 0.3 ± 0.6 | 0.7 ± 0.6 |
| mSA | 0.778 | 4.7 ± 2.1 | 2.3 ± 2.1 | 0.0 ± 0.0 | 0.0 ± 0.0 | 0.7 ± 0.6 | 2.0 ± 1.7 |
| SB | 0.353 | 2.3 ± 0.6 | 2.0 ± 0.0 | 0.0 ± 0.0 | 0.3 ± 0.6 | 0.0 ± 0.0 | 0.0 ± 0.0 |
| mSB | 1.569 | 3.3 ± 1.5 | 3.0 ± 2.0 | 0.0 ± 0.0 | 0.0 ± 0.0 | 0.0 ± 0.0 | 0.3 ± 0.6 |

[1] NC: negative control, R.D.W. 10% *v/v*; SA: sample A, 10% *v/v*; mSA: modified sample A, 10% *v/v*; SB, sample B, 10% *v/v*; mSB, modified sample B, 10% *v/v*. * Significant difference in comparison with the negative control and treated groups at *p* < 0.05.

### 3.3. MN Test

For the in vivo MN test, rats were treated with SA, mSA, SB, and mSB, and blood was obtained at 48 and 72 h after intratracheal instillation. At 48 h after administration, the RET ratio and the Mn-RET ratio of the SA, mSA, and mSB groups showed no significant difference compared to the negative control. The SB group showed a significantly decreased RET ratio but no significant difference in Mn-RET compared to the negative control. At 72 h after administration, the RET ratio and the Mn-RET ratio of the SA and mSA groups showed no significant difference compared to the negative control. The RET ratio of the SB and mSB groups was significantly decreased by 0.83 and 0.73 times that of the negative control (Table 8). The Mn-RET ratio of the SB group was significantly increased to 1.9 times that of the negative control. However, there was no significant difference between the mSB and negative control groups. The results of the SA, mSA, and mSB groups in the MN test were negative. However, since the Mn-RET ratio of the SB group was significantly increased by 1.9 times, but not a full twofold increase compared to the negative control group, the results of SB in the MN test were weakly positive.

**Table 8.** Changes of reticulocytes with micronuclei in the peripheral blood of rats after intratracheal instillation with unmodified and modified nano-silica in the peripheral blood micronucleus test.

| Group/ Intervals | RETs/NCEs [1] (‰) | Mn-RETs/RETs (‰) |
|---|---|---|
| 48 h | | |
| NC [2] | 37.2 ± 7.4 [3] | 8.1 ± 2.8 |
| PC | 6.2 ± 1.9 * | 62.1 ± 36.5 * |
| SA | 38.9 ± 7.0 | 7.5 ± 4.4 |
| mSA | 35.7 ± 13.2 | 6.9 ± 4.5 |
| SB | 25.7 ± 3.0 * | 12.4 ± 10.9 |
| mSB | 28.1 ± 5.0 | 7.1 ± 4.6 |
| 72 h | | |
| NC | 46.3 ± 8.2 | 4.6 ± 1.5 |
| PC | 19.2 ± 4.9 * | 8.0 ± 2.7 * |
| SA | 44.9 ± 6.6 | 4.2 ± 1.4 |
| mSA | 38.4 ± 12.1 | 8.0 ± 4.1 |
| SB | 34.9 ± 3.1 * | 8.8 ± 3.5 *,a |
| mSB | 33.6 ± 4.6 * | 4.7 ± 1.3 [a] |

[1] RETs: reticulocytes; NCEs: normochromatic erythrocytes; Mn-RETs: micronucleated reticulocytes. [2] NC: negative control, R.D.W, 0.5 mL/kg bw, i.t.; PC: positive control, cyclophosphamide, 20 mg/kg bw, 10 mL/kg bw, i.p.; SA: sample A, 0.645 mg/kg bw, 320× diluted, 0.5 mL/kg bw, i.t.; mSA: modified sample A, 0.06 mg/kg bw, 320× diluted, 0.5 mL/kg bw, i.t.; SB: sample B, 2.24 mg/kg bw, 80× diluted, 0.5 mL/kg bw, i.t.; mSB: modified sample B, 0.24 mg/kg bw, 80× diluted, 0.5 mL/kg bw, i.t. [3] Data are expressed as the mean ± SD (n = 4–5). * Significant difference in comparison with the negative control and treated groups at $p < 0.05$. [a] Significant difference in comparison with the SB and mSB groups at $p < 0.05$. Group: SA: sample A unmodified; mSA: modified sample A; SB: sample B; mSB: modified sample B.

### 3.4. Acute Pulmonary Toxicity Test (1 D)

The gross lesions of the lungs of the rats in negative control, SA, mSAH, mSAL, SB, and mSB groups were observed. Hemorrhage and emphysema in several lung lobes were noticed in all the treatment groups. The mSAH group showed most severe gross lesions and highest mortality (Figure 2, Table 9). Under histopathologic examination, the SA (Figure 3B), mSAL (Figure 3C), and SB (Figure 3D) groups showed higher pathological incidence in edema, necrosis, and inflammation and significantly higher total scores than the negative control (Figure 3A). The SB induced significantly more severe edema, inflammation, and necrosis than mSB (Figure 3E) in rat lungs. The lung weight might increase when the edema and inflammation in lung was induced by pulmonary damage (Figure 4, Table 10). The relative lung weight of rats in SA, mSAL, and SB groups was significantly increased compared to the negative control. In addition, the relative lung weight of the rats in SB group was significantly higher than in mSB group (Figure 4A). The total WBC, macrophage, neutrophil, and lymphocyte count of BALF in SA, mSAL, and SB groups were significantly increased compared to the negative control. It is noteworthy that the mSB did not significantly increase the inflammation cell count in the lung compared to the negative control, and the inflammation cell count was significantly lower than the SB group. On the other hand, in comparison with SA, the mSAL induced significantly more WBC accumulation in rat lungs (Table 11). The protein concentration of BALF in the SA and mSAL groups was significantly increased compared to the negative control (Figure 4B). The TNF-α concentration of BALF in the mSAL and SB groups was significantly increased compared to the negative control (Figure 4C).

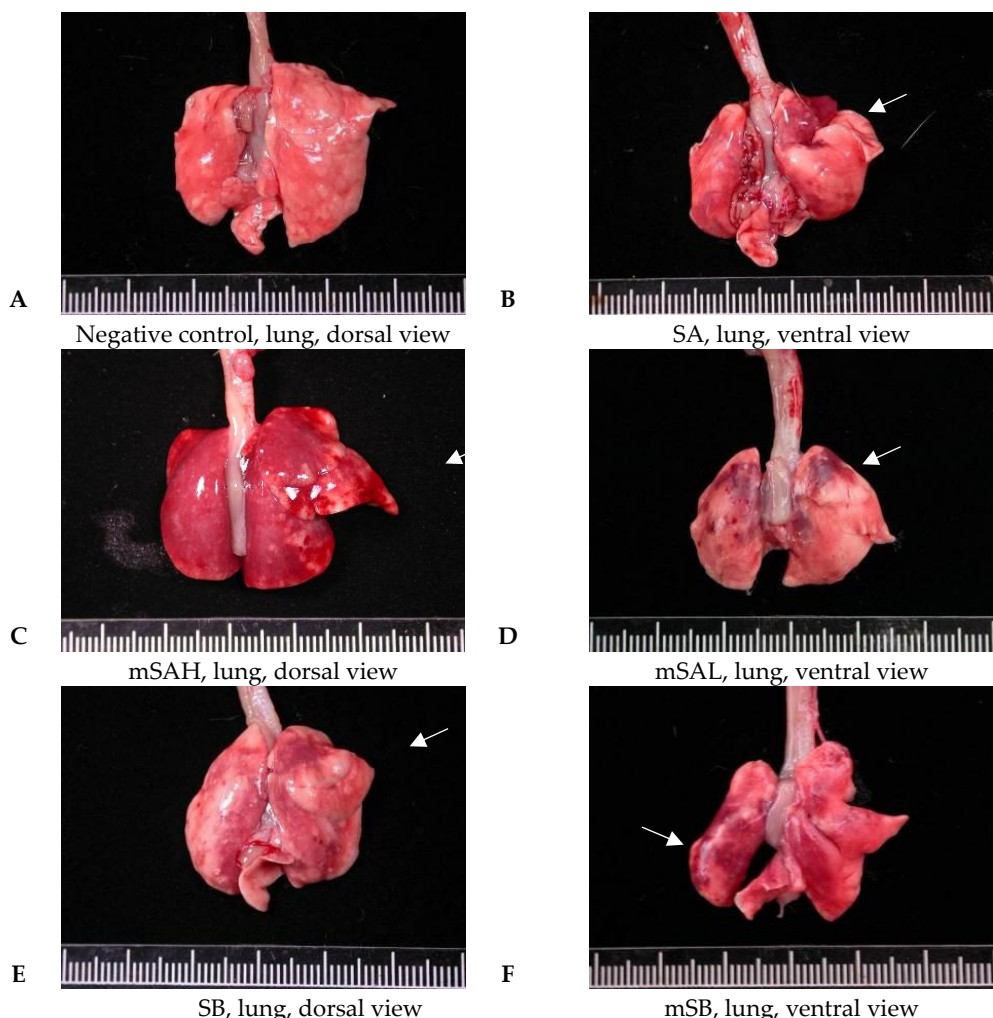

**Figure 2.** Gross lesion of lungs of rats after intratracheal instillation with unmodified and modified nano-silica in the acute pulmonary toxicity on Day 1. Gross lesions of lungs in negative control group (**A**), SA group (**B**), mSAH group (**C**), mSAL group (**D**), SB group (**E**), and mSB group (**F**) were found in rats at Day 1. Arrow.

**Table 9.** Clinical signs and time course of death of rats after intratracheal instillation with unmodified and modified nano-silica on Day 1.

| Group | Animal No. | Clinical Sign | Hours After Treatment | | | Mortality (%) [3] |
|---|---|---|---|---|---|---|
| | | | **1** | **12** | **24** | |
| NC [1] | 6 | Normal | 0 [2] | 0 | 0 | 0 |
| SA | 6 | Normal | 0 | 0 | 0 | 0 |
| mSAL | 5 | Normal | 0 | 0 | 0 | 0 |
| SB | 6 | Normal | 0 | 0 | 0 | 0 |
| mSB | 6 | Normal | 0 | 0 | 0 | 0 |
| mSAH | 3 | Dyspnea | 0 | 1 | 1 | 66.6 |

[1] NC: negative control, R.D.W, 0.5 mL/kg bw, i.t.; SA: sample A, 2.58 mg/kg bw, 80× diluted, 0.5 mL/kg bw, i.t.; mSAH: modified sample A high dose, 0.24 mg/kg bw, 80× diluted, 0.5 mL/kg bw, i.t.; mSAL: modified sample A low dose, 0.06 mg/kg bw, 320× diluted, 0.5 mL/kg bw, i.t.; SB: sample B, 2.24 mg/kg bw, 80× diluted, 0.5 mL/kg bw, i.t.; mSB: modified sample B, 0.24 mg/kg bw, 80× diluted, 0.5 mL/kg bw, i.t. [2] Number of rats observed. [3] Mortality (%) = (dead no./treated no.) × 100.

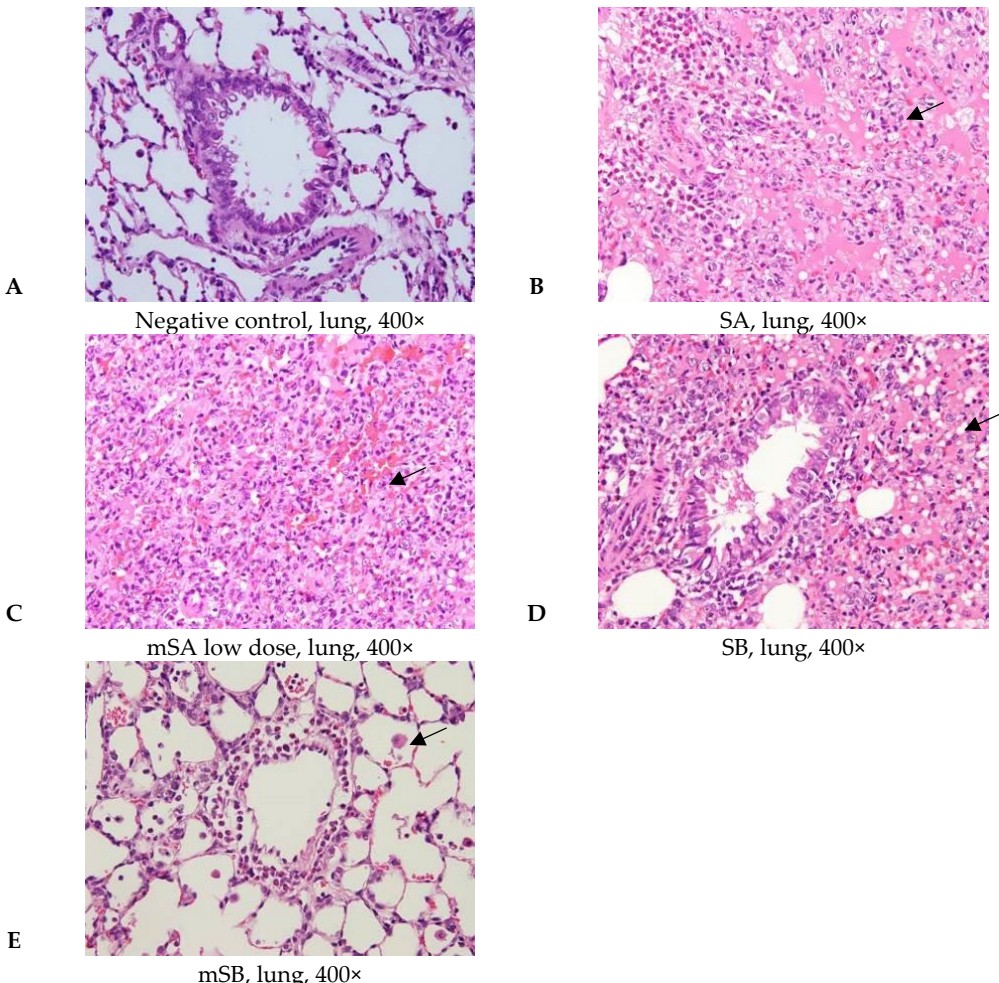

**Figure 3.** Histopathological changes of rats after intratracheal instillation with unmodified and modified nano-silica at Day 1 in the acute intratracheal instillation toxicity study. Normal architectures of lung in the control group (**A**), edema and inflammation of lung in the SA group (**B**), edema, hemorrhage, and inflammation of lung in the mSA low dose group (**C**), edema, hemorrhage, and inflammation of lung in the SB group (**D**), hemorrhage and inflammation of lung in mSB group (**E**) were found in rats at Day 1. H&E stain. Negative control: R.D.W, 0.5 mL/kg bw, i.t.; SA: sample A, 2.58 mg/kg bw, 80× diluted, 0.5 mL/kg bw, i.t.; mSAL: modified sample A low dose, 0.06 mg/kg bw, 320× diluted, 0.5 mL/kg bw, i.t.; SB: sample B, 2.24 mg/kg bw, 80× diluted, 0.5 mL/kg bw, i.t.; mSB: modified sample B, 0.24 mg/kg bw, 80× diluted, 0.5 mL/kg bw, i.t. H&E stain. 400×. Arrow.

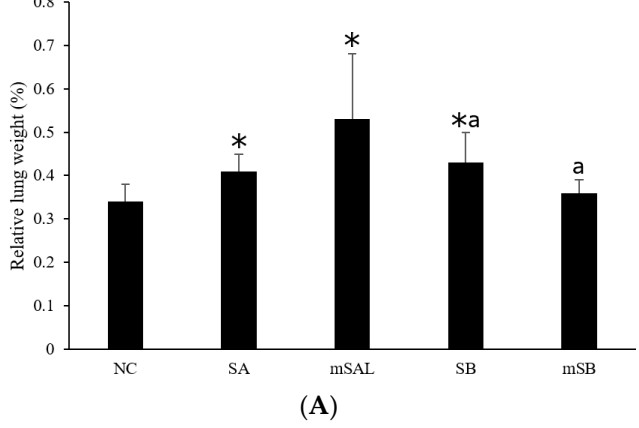

(**A**)

**Figure 4.** *Cont.*

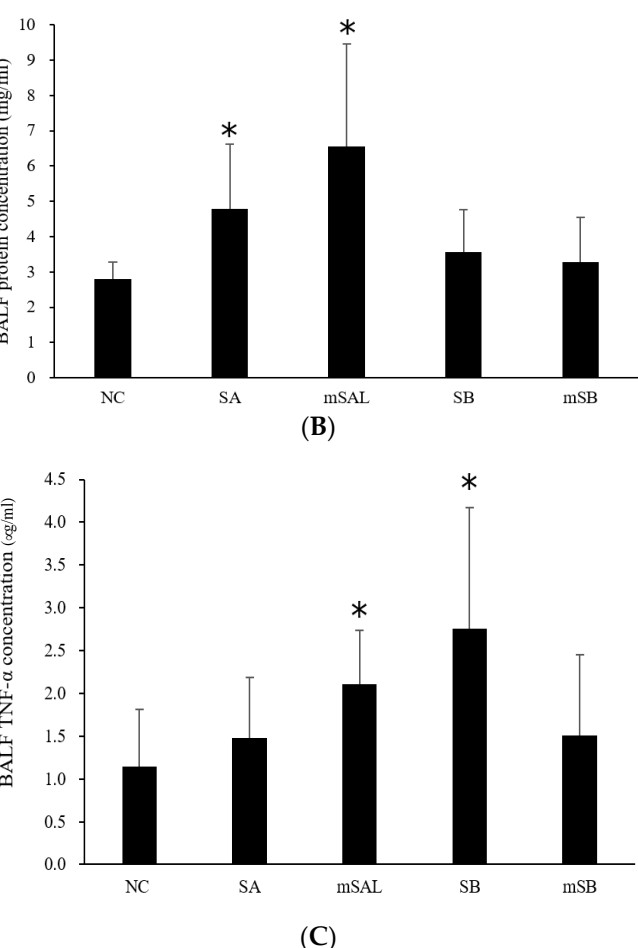

**Figure 4.** Relative lung weight, BALF protein concentration, and BALF TNF-$\alpha$ concentration of rats after intratracheal instillation with unmodified and modified nano-silica at Day 1 in the acute pulmonary toxicity study. Relative lung weights of SA, mSAL, and SB groups were significantly increased versus the control group (**A**), SB was significantly increased versus mSB group (**A**). BALF protein expression of SA and mSAL groups were significantly increased versus the control (**B**). BALF TNF-$\alpha$ concentrations of mSAL and SB groups were significantly increased versus the control group (**C**). NC: negative control, R.D.W, 0.5 mL/kg bw, i.t.; SA: sample A, 2.58 mg/kg bw, 80× diluted, 0.5 mL/kg bw, i.t.; mSAL: modified sample A low dose, 0.06 mg/kg bw, 320× diluted, 0.5 mL/kg bw, i.t.; SB: sample B, 2.24 mg/kg bw, 80× diluted, 0.5 mL/kg bw, i.t.; mSB: modified sample B, 0.24 mg/kg bw, 80× diluted, 0.5 mL/kg bw, i.t. * Significant difference between the control and treated groups at $p < 0.05$. [a] Significant difference between the SB and mSB groups at $p < 0.05$. Relative lung weight = lung weight (g)/body weight (g).

**Table 10.** Summary of pathological incidence and pathological scores of rats after intratracheal instillation with unmodified and modified nano-silica on Day 1.

| | | Pathological Incidence | | | | |
|---|---|---|---|---|---|---|
| | | Group | | | | |
| **Organ** | **Lesions** | **NC [1]** | **SA** | **mSAL** | **SB** | **mSB** |
| Lung | | | | | | |
| | Edema, extensive | - | 4/6 | 2/5 | 3/6 | - |
| | Hemorrhage, focal | - | - | 3/5 | 3/6 | 1/6 |
| | Inflammation, alveolar, focal | - | 6/6 | 4/5 | 6/6 | 3/6 |
| | Necrosis, alveolar wall, focal | - | 5/6 | 2/5 | 4/6 | - |

**Table 10.** *Cont.*

| Organ | Lesions | Group | | | | |
|---|---|---|---|---|---|---|
| | | NC | SA | mSAL | SB | mSB |
| Lung | | | | | | |
| | Edema, extensive | 0.0 ± 0.0 [2,3] | 1.8 ± 1.5 * | 1.0 ± 1.4 | 2.0 ± 2.2 *,[b] | 0.0 ± 0.0 [b] |
| | Hemorrhage, focal | 0.0 ± 0.0 | 0.0 ± 0.0 [a] | 1.2 ± 1.1*,[a] | 1.0 ± 1.1 * | 0.2 ± 0.4 |
| | Inflammation, alveolar, focal | 0.0 ± 0.0 | 2.7 ± 0.8 * | 2.0 ± 1.6 * | 3.5 ± 1.2 *,[b] | 0.7 ± 0.8 [b] |
| | Necrosis, alveolar wall, focal | 0.0 ± 0.0 | 2.2 ± 1.2 * | 1.0 ± 1.4 | 2.0 ± 1.7 *,[b] | 0.0 ± 0.0 [b] |
| | Total score [4] | 0.0 ± 0.0 | 6.7 ± 3.2 * | 5.2 ± 5.2 * | 8.5 ± 5.5 *,[b] | 0.8 ± 1.2 [b] |

[1] NC: negative control, R.D.W, 0.5 mL/kg bw, i.t.; SA: sample A, 2.58 mg/kg bw, 80× diluted, 0.5 mL/kg bw, i.t.; mSAL: modified sample A low dose, 0.06 mg/kg bw, 320× diluted, 0.5 mL/kg bw, i.t.; SB: sample B, 2.24 mg/kg bw, 80× diluted, 0.5 mL/kg bw, i.t.; mSB: modified sample B, 0.24 mg/kg bw, 80× diluted, 0.5 mL/kg bw, i.t. [2] Degree of lesions was graded from one to five depending on severity: 1 = minimal (<1%); 2 = slight (1–25%); 3 = moderate (26–50%); 4 = moderate/severe (51–75%); 5 = severe/high (76–100%). [3] Data are expressed as the mean ± SD (n = 5–6). [4] Total score = sum of all the pathological score of the left lobe of lung. * Significant difference between the control and treated groups at $p < 0.05$. [a] Significant difference between the SA and mSAL groups at $p < 0.05$. [b] Significant difference between the SB and mSB groups at $p < 0.05$.

**Table 11.** Changes of bronchoalveolar lavage fluid, WBC total count, and WBC differential count of rats after intratracheal instillation with unmodified and modified nano-silica on Day 1.

| Group | WBC [2] ($10^5$) | E ($10^5$) | N ($10^5$) | L ($10^5$) | M ($10^5$) |
|---|---|---|---|---|---|
| NC [1] | 6.1 ± 2.9 [3] | 0.0 ± 0.0 | 0.1 ± 0.1 | 0.3 ± 0.2 | 5.7 ± 2.7 |
| SA | 23.7 ± 12.2 *,[a] | 0.4 ± 1.0 | 10.1 ± 8.1 * | 1.1 ± 0.7 * | 12.2 ± 3.9 * |
| mSAL | 39.0 ± 10.8 *,[a] | 0.6 ± 0.7 | 20.4 ± 7.9 * | 2.0 ± 1.3 * | 16.1 ± 10.0 * |
| SB | 33.5 ± 19.2 *,[b] | 0.4 ± 0.4 * | 15.6 ± 14.0 *,[b] | 2.2 ± 1.4 *,[b] | 15.3 ± 6.5 * |
| mSB | 10.7 ± 10.7 [b] | 0.1 ± 0.2 | 0.8 ± 1.5 [b] | 0.5 ± 0.5 [b] | 9.4 ± 8.7 |

[1] NC: negative control, R.D.W, 0.5 mL/kg bw, i.t.; SA: sample A, 2.58 mg/kg bw, 80× diluted, 0.5 mL/kg bw, i.t.; mSAL: modified sample A low dose, 0.06 mg/kg bw, 320× diluted, 0.5 mL/kg bw, i.t.; SB: sample B, 2.24 mg/kg bw, 80× diluted, 0.5 mL/kg bw, i.t.; mSB: modified sample B, 0.24 mg/kg bw, 80× diluted, 0.5 mL/kg bw, i.t. [2] WBC, white blood cell; E, eosinophil; N, neutrophil; L, lymphocyte; M, macrophage. [3] Mean of data is expressed as the mean ± SD (n = 5–6). * Significant difference between the control and treated groups at $p < 0.05$. [a] Significant difference between the SA and mSAL groups at $p < 0.05$. [b] Significant difference between the SB and mSB groups at $p < 0.05$.

### 3.5. Acute Pulmonary Toxicity Test (14 D)

There was no significant gross lesion in the lungs of all the treatment groups on Day 14 (Figure 5). However, in histopathologic examination, minimal to moderate macrophage aggregations were observed in the lungs of the SA (Figure 6B), mSA (Figure 6C), and SB (Figure 6D) groups at high incidence. In the pathologic scores of macrophage aggregations and total scores, the SA, mSA, and SB groups were significantly increased compared to the negative control (Figure 6A). In addition to macrophage aggregations, minimal to slight alveolar granulomatous inflammation was noticed in the lungs of SB group. It is noteworthy that mSB (Figure 6E) induced no significant lesions, and all the pathologic scores of lungs were significantly lower than the SB group (Figure 6, Table 12). The neutrophil count of BALF of SB group was significantly increased compared to the negative control (Table 13).

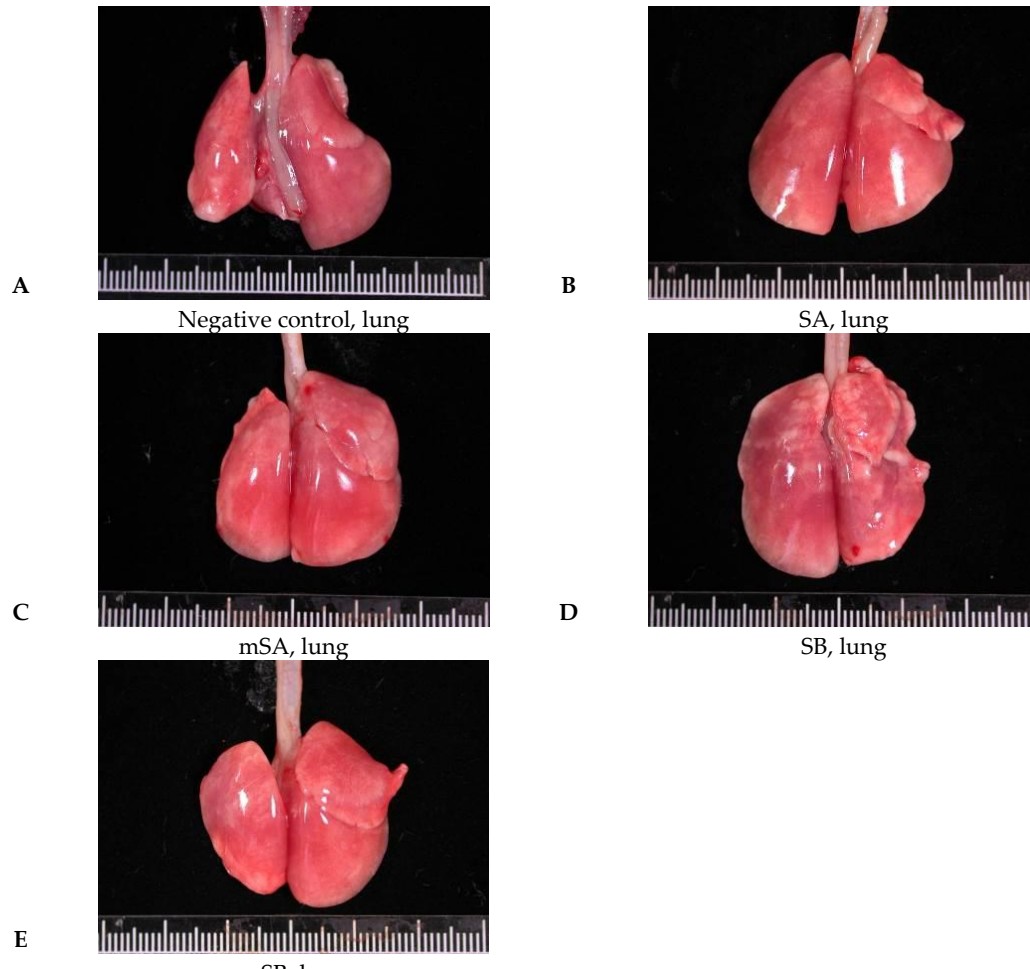

**A** Negative control, lung

**B** SA, lung

**C** mSA, lung

**D** SB, lung

**E** mSB, lung

**Figure 5.** Gross findings of lungs of rats after intratracheal instillation with unmodified and modified nano-silica in the acute pulmonary toxicity on Day 14. Normal gross of lung of control group (**A**), SA group (**B**), mSA group (**C**), SB group (**D**), and mSB group (**E**) were found in rats. Negative control: R.D.W, 0.5 mL/kg bw, i.t.; SA: sample A, 0.645 mg/kg bw, 320× diluted, 0.5 mL/kg bw, i.t.; mSA: modified sample, 0.06 mg/kg bw, 320× diluted, 0.5 mL/kg bw, i.t.; SB: sample B, 2.24 mg/kg bw, 80× diluted, 0.5 mL/kg bw, i.t.; mSB: modified sample B, 0.24 mg/kg bw, 80× diluted, 0.5 mL/kg bw, i.t.

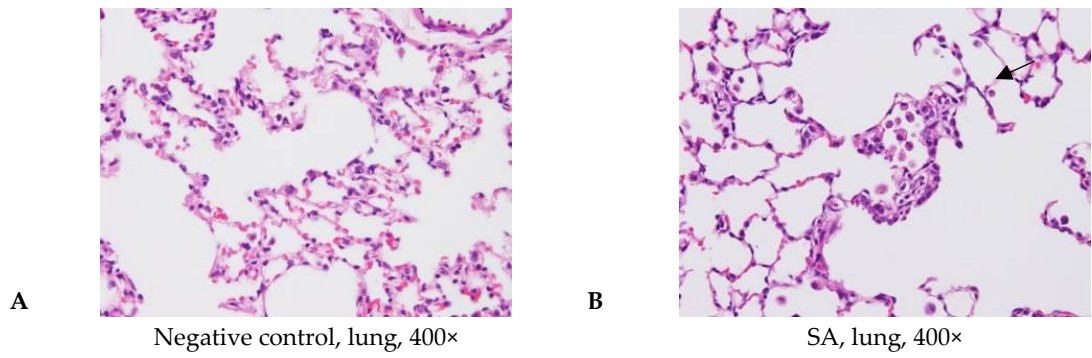

**A** Negative control, lung, 400×

**B** SA, lung, 400×

**Figure 6.** *Cont.*

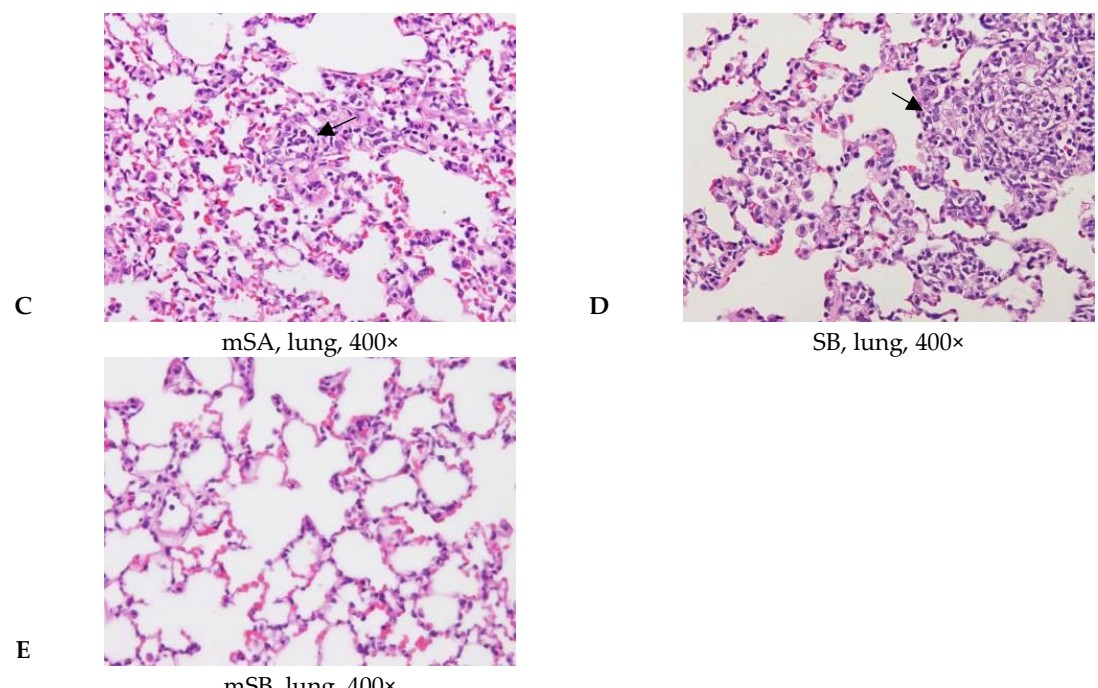

**Figure 6.** Histopathologic findings of lungs of rats after intratracheal instillation with unmodified and modified nano-silica in the acute pulmonary toxicity study (14 D). Normal architectures of lung in the control group (**A**) and mSB group (**E**), minimal to slight macrophage aggregations of the lung of SA and mSA groups (**B**,**C**), and slight to moderate macrophage aggregations with minimal to slight granulomatous inflammation of the lung of SB group (**D**) were found in rats. Negative control: R.D.W, 0.5 mL/kg bw, i.t.; SA: sample A, 0.645 mg/kg bw, 320× diluted, 0.5 mL/kg bw, i.t.; mSA: modified sample, 0.06 mg/kg bw, 320× diluted, 0.5 mL/kg bw, i.t.; SB: sample B, 2.24 mg/kg bw, 80× diluted, 0.5 mL/kg bw, i.t.; mSB: modified sample B, 0.24 mg/kg bw, 80× diluted, 0.5 mL/kg bw, i.t. H&E stain. 400×. Arrow.

**Table 12.** Summary of pathological incidence and pathological scores of rats after intratracheal instilled with unmodified and modified nano silica on Day 14.

| Pathological Incidence | | | | | | |
|---|---|---|---|---|---|---|
| **Organ** | **Lesions** | **Group** | | | | |
| | | **NC** [1] | **SA** | **mSA** | **SB** | **mSB** |
| Lung | | | | | | |
| | Aggregate of macrophage, alveolar multifocal | - | 4/5 [2] | 5/5 | 5/5 | 0/5 |
| | Granulomatous inflammation, alveolar, focal | - | - | - | 5/5 | 0/5 |
| Pathological Scores | | | | | | |
| **Organ** | **Lesions** | **Group** | | | | |
| | | **NC** | **SA** | **mSA** | **SB** | **mSB** |
| Lung, left lobe | | | | | | |
| | Aggregate of macrophage, alveolar multifocal | 0.0 ± 0.0 [3,4] | 2.2 ± 1.5 * | 1.6 ± 0.5 * | 2.2 ± 0.4 *,a | 0.0 ± 0.0 a |
| | Granulomatous inflammation, alveolar, focal | 0.0 ± 0.0 | 0.0 ± 0.0 | 0.0 ± 0.0 | 1.6 ± 0.5 *,a | 0.0 ± 0.0 a |
| | Total score [5] | 0.0 ± 0.0 | 2.2 ± 1.5 * | 1.6 ± 0.5 * | 3.8 ± 0.8 *,a | 0.0 ± 0.0 a |

[1] NC: negative control, R.D.W, 0.5 mL/kg bw, i.t.; SA: sample A, 0.645 mg/kg bw, 320× diluted, 0.5 mL/kg bw, i.t.; mSA: modified sample, 0.06 mg/kg bw, 320× diluted, 0.5 mL/kg bw, i.t.; SB: sample B, 2.24 mg/kg bw, 80× diluted, 0.5 mL/kg bw, i.t.; mSB: modified sample B, 0.24 mg/kg bw, 80× diluted, 0.5 mL/kg bw, i.t. -: No significant lesion. [2] Incidence: affected rats/total examined rats (n = 5). [3] Degree of lesions was graded from one to five depending on severity: 1 = minimal (<1%); 2 = slight (1–25%); 3 = moderate (26–50%); 4 = moderate/severe (51–75%); 5 = severe/high (76–100%). [4] Data are expressed as the mean ± SD (n = 5). [5] Total score = sum of all the pathological score of the left lobe of lung. * Significant difference between the control and treated groups at $p < 0.05$. [a] Significant difference between the SB and mSB groups at $p < 0.05$.

**Table 13.** Changes of bronchoalveolar lavage fluid, WBC total count, and WBC differential count of rats after intratracheal instillation with unmodified and modified nano-silica on Day 14.

| Group | M ($10^5$) | N ($10^5$) | L ($10^5$) | E ($10^5$) |
|---|---|---|---|---|
| NC [1] | 3.41 ± 0.88 [2] | 0.03 ± 0.01 | 0.07 ± 0.05 | 0.00 ± 0.00 |
| SA | 3.94 ± 0.49 | 0.18 ± 0.16 | 0.16 ± 0.08 | 0.01 ± 0.03 |
| mSA | 7.28 ± 4.11 | 0.03 ± 0.03 | 0.19 ± 0.13 | 0.00 ± 0.00 |
| SB | 5.09 ± 2.97 | 0.29 ± 0.24 * | 0.39 ± 0.41 | 0.00 ± 0.00 |
| mSB | 3.97 ± 2.31 | 0.23 ± 0.34 | 0.16 ± 0.07 | 0.00 ± 0.00 |

[1] NC: negative control, R.D.W, 0.5 mL/kg bw, i.t.; SA: sample A, 0.645 mg/kg bw, 320× diluted, 0.5 mL/kg bw, i.t.; mSA: modified sample, 0.06 mg/kg bw, 320× diluted, 0.5 mL/kg bw, i.t.; SB: sample B, 2.24 mg/kg bw, 80× diluted, 0.5 mL/kg bw, i.t.; mSB: modified sample B, 0.24 mg/kg bw, 80× diluted, 0.5 mL/kg bw, i.t. [2] Data are expressed as the mean ± SD (n = 5). * Significant difference between the control and treated groups at $p < 0.05$.

## 4. Discussion

Nano-silicas are one of the most commonly used nano-additives in many products. Silicas are divided into three major forms: crystalline, cryptocrystalline, and amorphous. Crystalline silicas can be further divided into three crystalline forms, such as quartz, cristobalite, and tridymite. Quartz is one of the most abundant minerals in the Earth's crust and is, therefore, also the most common dust that miners are exposed to [27]. Long-term exposure to silica dust may cause silica to be deposited in the alveolar space and induce silicosis or lung cancer. Silicosis may be further classified as simple (nodular) silicosis, acute silicosis, and complex pneumoconiosis [28]. In addition to silicosis, silicas may also induce many autoimmune diseases, such as systemic sclerosis, rheumatoid arthritis, systemic lupus erythematosus, etc. [29]. When compared to amorphous silicas, crystalline silicas show higher cytotoxicity due to irregular particles with sharp edges, stable surface radicals, and sustained release of HO radicals via a Fenton-like mechanism [30].

To compare the genotoxicity of the unmodified and modified silica samples, we conducted an Ames test, chromosomal aberration test, and micronucleus test. The Ames test is a quick and simple genotoxicity test conducted with mutated *Salmonella typhimurium* strains. The TA98 strain has an allele of hisD3052, which is sensitive to frameshift mutations. On the other hand, the TA100 strain has an allele of hisG46, which is sensitive to base-pair substitution mutations of the C–G base pair [31]. In our study, SB induced bacterial reverse mutation in the TA100 strain but not in the TA 98 strain. The positive result in TA100 and the negative result in TA98 in the Ames test indicated that SB might induce base-pair substitution mutations, not frameshift mutations, in *Salmonella typhimurium*. Some reports have indicated that no mutagenic potential is observed in the Ames test with or without metabolic activation [32–34]. One study showed the genotoxic potential of exfoliated silicate nanoclay [35].

The chromosomal aberration test is an in vitro genotoxicity test conducted in a CHO-K1 cell line. In this test, the samples were tested to see if they could induce chromosome recombination, breaks, polyploidy, or other morphological abnormalities in rapidly dividing mammal cells. These morphological abnormalities are also called chromosome aberrations. To induce chromosome aberrations, at least one double-strand-break (DSB) of DNA is required [36]. Chromosome aberrations can be further divided into chromosome-type aberrations (CSAs) and chromatid-type aberrations (CTAs) according to the mechanisms that induce the chromosome aberration. CSAs are mainly induced by ionizing radiation. On the other hand, CTAs are mainly produced by chemical injury [37,38]. In a previous study, the CA test of the nano-silica in CHO-K1 cells showed that nano-silica did not cause the ratio of chromosomal aberrations to increase with or without S9 metabolism [39]. However, in our study, SB induced significant increases in the frequency of chromosome aberrations. On the other hand, the major type of chromosome aberration that SB induced was gaps, which belong to CSAs. The results of the CA test indicated that SB may induce ionizing radiation damage to the genes of CHO-K1 cells.

The micronucleus test is an in vivo genotoxicity test conducted by observing the frequency of micronuclei in the peripheral blood of rodents. The majority of mature RBCs

of mature rodents differentiate and proliferate from hematopoietic stem cells in the bone marrow. During hematopoietic stem cell differentiation and proliferation, the test articles may induce chromosomal damage in the stem cells. The damaged chromosome would not be able to attach onto spindle fibers and remain in the plasma. When normoblasts differentiate into polychromatic erythrocytes (PCE), the nucleus can be ejected. The damaged chromosome, which remains in the plasma, is called a micronucleus. This test can simultaneously test whether the test articles would cause clastogenicity or aneugenicity [40]. In a previous study, the MN test in ICR mice at 48 h after silica oral administration showed negative results [38]. In addition to oral administration, another study showed that rats which were given an intratracheal instillation of nano-silica did not present an increase in the Mn-RET ration in bone marrow at 72 h after administration [10]. However, Wistar rats which were intravenously injected with 15 and 55 nm silica showed positive results in the MN test [8]. The results indicate that different routes of administration might affect the genotoxicity of silica in the MN test. The data shown in the table are the ratio of RETs/NCEs and Mn-RETs/RETs, not the absolute count of Mn-RET. After being released into the peripheral blood circulation, the RET will mature and become NCE within 48 h. Therefore, the Mn-RETs we noticed at the 48 h timepoint may become Mn-NCEs at the 72 h timepoint. Also, RETs/NCEs at the 72 h timepoint were higher than 48 h timepoint. This is because the inhibition of RET production caused by the test article/positive control toxicity was declined after 48 h. Then, the bounced RET production was noticed at the 72 h timepoint. In summary, the decreased Mn-RETs/RETs at the 72 h timepoint was caused by (1) the Mn-RET noticed at 48 h timepoint maturing and becoming "Mn-NCEs" which were not counted as Mn-RETs at the 72 h timepoint and (2) the Mn-RETs maybe being diluted by the increased RET production noticed at the 72 h timepoint.

To summarize the results of the genotoxicity tests, SB made the chromosomal aberration frequency of CHO-K1 cells significantly higher than the negative control. Though the SB group showed positive results in the Ames test and CA test, the genotoxicity of SB was reversed to produce negative results in these two tests after being metabolized with rat liver S9 extract. In the MN test, SB induced a weak positive result, which indicated that SB might cause gene damage in rodent hematopoietic cells. SB produced weakly positive or positive results in these genotoxicity tests. However, genotoxicity may be reduced after S9 metabolism or modification, which suggests that the genotoxicity of SB may be reduced by liver metabolism. In contrast, mSB did not induce significant genotoxicity in any of the three genotoxicity tests. The opposite result between SB and mSB indicated that the modification of SB might reduce genotoxicity.

In the acute pulmonary toxicity test, the rats were given an intratracheal instillation of diluted samples and sacrificed after 1 or 14 days. High mortality in the mSA rats was observed at 1 day after IT. In contrast, the other samples did not induce death in rats. Histopathologic examination of the lungs of the SA, mSAL, and SB groups showed remarkable edema, hemorrhage, inflammation, and alveolar wall necrosis. Other pulmonary toxicity parameters that can represent pulmonary toxicity of samples in rats, such as relative lung weight, BALF WBC count, BALF neutrophil count, BALF protein concentration, and BALF TNF-$\alpha$ concentration, showed similar results in the SA, mSAL, and SB groups. However, the SB group showed slight pulmonary damage in all of these parameters. After 14 days, aggregates of macrophages and granulomatous inflammation, which represent subacute pulmonary inflammation, were noticed in the lungs of the SA, mSA, and SB groups but not in the mSB group. The total histopathologic scores of the mSB group were significantly lower than the SB group at both 1 and 14 days, but there was no significant difference between the SA and mSA (low dose) groups. In summary, the modification of SA may increase pulmonary toxicity in rats 24 h after IT, based on the high mortality observed in the mSAH group, but there was no significant difference observed after 14 days. In addition, the modification of SB may reduce pulmonary toxicity in rats. According to the laws in Taiwan, the maximum permissible exposure dose of pure silica dust is 0.098 mg/m$^3$

(Ministry of Labor, Executive Yuan, Taiwan, 2010). The exposure dose of rats was calculated using the formula below:

Exposure concentration of nanoparticle (0.098 mg/m$^3$) × tidal volume of rat (2.1 mL) × breathing frequency in rat (102 times/min) × exposure hours in day (8 h) × particle deposition efficiency (10%)/rat body weight (300 g) = the exposure dose of rat (0.0034 mg/kg) [9].

The doses of SA, mSAH, mSAL, SB, and mSB in the acute pulmonary toxicity test (1 D) were 758.8, 70.5, 17.6, 658.8, and 70.5 times the maximum permissible exposure doses, respectively. On the other hand, the doses of SA, mSA, SB, and mSB in the acute pulmonary toxicity test (14 D) were 13.6, 1.3, 47.0, and 5.0 times the maximum permissible exposure doses, respectively. The test articles in the pulmonary toxicity tests might induce severe pulmonary toxicity, but the dose used in the tests was several times higher than the maximum permissible exposure dose. Without a value for the no-observed-adverse-effect level (NOEAL), we cannot conclude that the silica samples which were used in the tests are harmful to humans. However, the relative pulmonary toxicity between all four samples was confirmed.

In our research, the trifluorosilane surface modification may reduce the genotoxicity in the Ames test, chromosomal aberration test (CA), and rat peripheral blood micronucleus test (MN), as well as pulmonary toxicity in rats.

## 5. Conclusions

In conclusion, the genotoxicity and pulmonary toxicity of SB was significantly reduced after modification. On the other hand, the SA and mSA groups showed no genotoxicity. The modification of SA might increase pulmonary toxicity in rats 1 day after IT, but no significant difference of pulmonary toxicity in rats was noticed after 14 days.

**Author Contributions:** Y.-T.C., P.-J.C. and J.-W.L. designed the study. P.-Y.L. and P.-W.C. performed the experiments and analyzed the data. Y.-T.C., F.-J.T. and J.-W.L. wrote the first draft of the manuscript. All authors contributed to the interpretation of the data and critical revision the manuscript. All authors have read and agreed to the published version of the manuscript.

**Funding:** This research received a Ministry of Science and Technology (MOST) Taiwan grant (MOST 104-2621-M-005-001).

**Institutional Review Board Statement:** The animal experiments were fulfilled according to the Guide for the Care and Use of Laboratory Animals. All protocols were approved by the Institutional Animal Care and Use Committee (IACUC) of the National Chung Hsing University in 2015, Taiwan (IACUC:104-136).

**Informed Consent Statement:** Not applicable.

**Data Availability Statement:** Not applicable.

**Conflicts of Interest:** The authors declare no conflict of interest.

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
