# Peer review of "Comparison of Genotoxicity and Pulmonary Toxicity Study of Modified SiO2 Nanomaterials"

_applsci, doi:10.3390/app112411990_

Round 1
Reviewer 1 Report
In this article the authors study the genotoxicity and the acute pulmunary toxicity of four samples of nano SiO2: two pristine and two modified.
I suggest to increase the number of reference in the introduction, because the toxicity of the silica is one on the main topics of the last twenty years.
The mechanisms of toxicity were studied and well characterised during the time and severals articles could be cited.
Finally just few question on the samples characterization:
How you calculate the measures of the single particles? From TEM image? How many particles they measured?
The TEM images show a very non homogeneus samples in particular sample B.
Author Response
Reviewer 1
In this article the authors study the genotoxicity and the acute pulmunary toxicity of four samples of nano SiO2: two pristine and two modified.
I suggest to increase the number of reference in the introduction, because the toxicity of the silica is one on the main topics of the last twenty years.
Response:
Thank you very much for your professional and useful comments. According to your suggestion, we added the number of references in the introduction. (reference 3, 5, 11-21)
The mechanisms of toxicity were studied and well characterized during the time and severals articles could be cited.
Response:
We have cited several articles regarding mechanisms of toxicity. (reference 15-21, 32-35)
Finally just few question on the samples characterization:
How you calculate the measures of the single particles? From TEM image? How many particles they measured?
Response:
The structure of test particles was examined using a transmission electronic microscopy. After diluting in purified water, samples were placed over a copper grid coated with carbon film followed by staining with 2% phosphotungstic acid. The samples were air dried prior to placement in the TEM instrument for analysis. The shapes and sized of nano silicas were determined using a TEM, which showed that particles diameters in four different preparations were in the nanosized range (1 – 100 nm). The average particle sizes of SA, mSA, SB, mSB are 55.16, 82.29, 22, and 86.64 nm and the solid content are 41.13,3.77,35.8,3.77% respectively. We added these descript on “2.1 Sample” section.
The TEM images show a very non homogeneus samples in particular sample B.
Response:
We agree that the sizes of mSA determined by TEM appear to be less homogeneous shown in Fig. 1B, and this could have resulted from incomplete modification of the particles.
Reviewer 2 Report
The manuscript is devoted to the toxicity of silica-based nanomaterials. In general, a lot of work has been done, many results have been obtained, however, the presentation of these results in the text of the manuscript is not always done in the best way. Further, I took the liberty of mentioning a number of suggestions for improving the text of the manuscript:
Conceptual question. Why are the authors investigating just such silica preparations? How were these nanomaterials obtained? What is their stability?
- Nanosized silica under the action of visible light can lead to the generation of reactive oxygen species. This is what may underlie toxicity. What do the authors know about this? Are there any data on the preparations used by the authors?
- Table 1 contains the TEOS abbreviation. Nowhere in the text of the manuscript is there such an abbreviation. It is necessary to decrypt! In general, in table 1 there is an unsigned column in which the numbers are written 1 1 2 1 1 2. The column should also be marked!
- How the authors explain the different dilutions of their preparations. Sometimes authors use SA 80 × diluted, sometimes SA 320 × diluted. Why?
- Does it make sense to show mean ± SD at (n = 3)? Maybe you need to use mean ± SEM?
- Table 5 (mutagenicity test). What is the difference between the first and second parts of the table, apart from positive control? Why is there such a huge spread of data in the same groups?
- When conducting a micronucleus test on the red bone marrow, the maximum concentration of micronuclei is observed within 24-28 hours after exposure. With an increase in the time after exposure, the effect decreases, since reticulocytes are released into the peripheral blood. Reminder, the erythrocyte lifespan is 120 days. That is, in the peripheral blood, micronuclei should only accumulate over time. From the data presented in table 8, it is obvious that after 72 hours there are fewer micronuclei in the peripheral blood. How can this be?
- It is not at all clear from the text of the manuscript how are representatives the data presented in Figure 3. The same applies to the next figure on histology.
- I'm not a native speaker of English, but even I think that the style of the article needs to be adjusted.
Author Response
The manuscript is devoted to the toxicity of silica-based nanomaterials. In general, a lot of work has been done, many results have been obtained, however, the presentation of these results in the text of the manuscript is not always done in the best way. Further, I took the liberty of mentioning a number of suggestions for improving the text of the manuscript:
Conceptual question. Why are the authors investigating just such silica preparations? How were these nanomaterials obtained? What is their stability?
Response:
Thank you very much for your professional and useful comments. According to your suggestion, we revised our manuscript.
We have studied the biological effects of different nanoparticles for the past few years and reported their molecular events underpinning their cytotoxicity (references below). However, we have not yet evaluated the effect of nanoparticles with in vivo studies. Due to the high number of poorly water-soluble active pharmaceutical ingredients, oral drug delivery development has become challenging. One of the strategies to enhance drug solubility and to achieve high oral bioavailability is to formulate such compounds into amorphous solid dispersions. In recent years, porous materials have been investigated as possible carriers into which a drug can be adsorbed, such as mesoporous silica, in particular. Unlike the ordered mesoporous network of silica, non-ordered silica already has a "generally regarded as safe" status, and is already used as an excipient in pharmaceutical and cosmetic products. Thus, it is reasonable to expect that products that contain solid dispersions with non-ordered carriers will reach the market sooner and more easily than those with ordered mesoporous carriers. With the collaboration with Industrial Technology Research Institute, we carried the animal studies to understand the genotoxicity and pulmonary toxicity of SiO2 nanomaterials.
- Lee, Y.H.,Chuang, S.M., Huang, S.C., Tan, X., Liang, R.Y., Yang, G.C., Chueh, P.J.* (2017) Biocompatibility assessment of nanomaterials for environmental safety screening. Environmental Toxicology 32(4):1170-1182. doi: 10.1002/tox.22313
- Chueh, P.J., Liang, R.Y., Lee, Y.H., Zeng, Z.M., Chuang, S.M.* (2014) Differential cytotoxic effects of gold nanoparticles in different mammalian cell lines. J Hazard Mater. 264, 303-312.
- Chuang SM, Lee YH, Liang RY, Roam GD, Zeng ZM, Tu HF, Wang SK, Chueh, P.J.* (2013) Extensive evaluations of the cytotoxic effects of gold nanoparticles. Biochim Biophys Acta. 1830(10):4960-73.
- Huang, S., Chueh, P.J., Lin, Y.W., Shih, T.S., Chuang, S.M. (2009) Disturbed mitotic progression and genome segregation are involved in cell transformation mediated by nano-TiO2 long-exposure. Toxicol Appl Pharmacol. 241(2):182-94.
- Hsin, Y.H., Chen, C.F., Huang, S., Shih, T.S., Lai, P.S., Chueh, P.J.* (2008) The apoptotic effect of nanosilver is mediated by a ROS- and JNK-dependent mechanism involving the mitochondrial pathway in NIH3T3 cells. Toxicol. Lett. 179(3), 130-139.
- Nanosized silica under the action of visible light can lead to the generation of reactive oxygen species. This is what may underlie toxicity. What do the authors know about this? Are there any data on the preparations used by the authors?
Response:
All the silica nanoparticle samples were sealed and kept away (in dark) from light.
- Table 1 contains the TEOS abbreviation. Nowhere in the text of the manuscript is there such an abbreviation. It is necessary to decrypt! In general, in table 1 there is an unsigned column in which the numbers are written 1 1 2 1 1 2. The column should also be marked!
Response:
We added the note under Table 1, TOES: tetraethoxysilane. We removed column 1 1 2 1 1 2 in Table 1.
- How the authors explain the different dilutions of their preparations. Sometimes authors use SA 80 × diluted, sometimes SA 320 × diluted. Why?
Response:
High mortality within 24 hours after i.t. was noticed in the mSAH group animal in the acute pulmonary toxicity test (1 D). Therefore, in the rat peripheral blood micronucleus test and acute pulmonary toxicity test (14 D), to ensure all the animals would survive to the end of the study, the dose level of mSA group was reduced to 320x. Then, to be comparable with the mSA group, the dose level of SA group was also reduced to 320x.
- Does it make sense to show mean ± SD at (n = 3)? Maybe you need to use mean ± SEM?
Response:
Use the standard error of the mean (SEM) instead of the standard deviation (SD) to represent the discrete program of the data. The SEM must be less than the SD value because the SD value divided by the square root of the number of samples is equal to the SEM. SEM is not used to express the degree of dispersion of measurement variables. It is used to indicate the accuracy of the sample average. In scientific papers, SD is used to indicate the degree of dispersion of data. The standard deviation is not just comparable to the sample average to be calculated. Other statistics (Statistics), such as odds ratio (OR), relative risk, and percentage, can also be used to calculate the SD value.
- Table 5 (mutagenicity test). What is the difference between the first and second parts of the table, apart from positive control? Why is there such a huge spread of data in the same groups?
Response:
The first part experiments were without S9, second part experiments were with S9 activation.
- When conducting a micronucleus test on the red bone marrow, the maximum concentration of micronuclei is observed within 24-28 hours after exposure. With an increase in the time after exposure, the effect decreases, since reticulocytes are released into the peripheral blood. Reminder, the erythrocyte lifespan is 120 days. That is, in the peripheral blood, micronuclei should only accumulate over time. From the data presented in table 8, it is obvious that after 72 hours there are fewer micronuclei in the peripheral blood. How can this be?
Response:
The data shown in the table are the ratio of RETs/NCEs and Mn-RETs/RETs, not the absolute count of Mn-RET. After being released to the peripheral blood circulation, the RET will mature and become NCE within 48 hours. Therefore, the Mn-RETs we noticed at the 48 hours timepoint may become Mn-NCEs at the 72 hours timepoint. Also, RETs/NCEs at the 72 hours timepoint was higher than 48 hours timepoint. This is because the inhibition of RET production caused by the test article/positive control toxicity was declined after 48 hours. Then, the bounced RET production was noticed at the 72 hours timepoint. In summary, the decreased Mn-RETs/RETs at the 72 hours timepoint was caused by 1) the Mn-RET noticed at 48 hours timepoint matured and become "Mn-NCEs" which was not counted as Mn-RETs at the 72 hours timepoint and 2) the Mn-RETs may be diluted by the increased RET production noticed at the 72 hours timepoint.
- It is not at all clear from the text of the manuscript how are representatives the data presented in Figure 3. The same applies to the next figure on histology.
Response:
We revised the Figure 3 descript in the text: In the histopathologic examination, the SA (Fig. 3B), mSAL (Fig 3C), SB (Fig 3D) groups showed higher pathological incidence in edema, necrosis, and inflammation and significantly higher total scores than the negative control (Fig. 3A). The SB induced significantly more severe edema, inflammation, and necrosis than mSB (Fig. 3E) in rat lungs.
We revised the Figure 5 descript in the text: However, in the histopathologic examination, minimal to moderate, macrophage aggregations were observed in the lungs of SA (Fig. 6B), mSA (Fig. 6C), and SB (Fig. 6D) groups at high incidence. In the pathologic scores of macrophage aggregations and total scores, the SA, mSA, and SB groups were significantly increased compared to the negative control (Fig. 6A). In addition to macrophage aggregations, minimal to slight alveolar granulomatous inflammation was noticed in the lungs of SB group. Noteworthy, mSB (Fig. 6E) induced no significant lesion, and all the pathologic scores of lungs were significantly lower than SB group (Fig. 6, Table 12).
- I'm not a native speaker of English, but even I think that the style of the article needs to be adjusted.
Response:
Our manuscript has been revised by a native English speaker.
Reviewer 3 Report
The manuscript entitled “Comparison of Genotoxicity and Pulmonary Toxicity Study of Modified SiO2 Nanomaterials” aimed to investigate genotoxicity and acute pulmonary toxicity of nano SiO2 with or without modification. The obtained results revealed that most of the tested samples showed negative results in all genotoxicity tests, except one that showed weakly positive reaction in chosen assays, but the genotoxicity could be reversed after the S9 metabolism or the modification.
Last sentence in the Introduction section is more of description of the results and should be removed and/or placed in Results or Discussion section.
I would suggest to include a reference for conducting CA analysis.
In Figure 4, please use larger font on x and y axes.
In the Discussion section, Authors could connect their results with the existing results on SiO2 nanoparticles genotoxicity.
Minor remarks:
In SiO2, 2 should be in subscript.
I would suggest using h/min instead of hours/minutes.
Page 4, line 188 – change to mammalian cell line
Put Salmonella Typhimurium in italics.
Author Response
The manuscript entitled “Comparison of Genotoxicity and Pulmonary Toxicity Study of Modified SiO2 Nanomaterials” aimed to investigate genotoxicity and acute pulmonary toxicity of nano SiO2 with or without modification. The obtained results revealed that most of the tested samples showed negative results in all genotoxicity tests, except one that showed weakly positive reaction in chosen assays, but the genotoxicity could be reversed after the S9 metabolism or the modification.
Last sentence in the Introduction section is more of description of the results and should be removed and/or placed in Results or Discussion section.
Response:
Thank you very much for your professional and useful comments.
According to your suggestion, we removed the last sentence in the introduction section to the discussion section.
I would suggest including a reference for conducting CA analysis.
Response:
We have added the reference:
- Organization for Economic Co-operation and Development (OECD). 1997. In vitro mammalian chromosome aberration test. In OECD Guidelines for the Testing of Chemicals. OECD Paris, Test Guideline No: 473. pp 1-10.
In Figure 4, please use larger font on x and y axes.
Response:
We use a larger font on x and y axes in Figure 4.
In the Discussion section, Authors could connect their results with the existing results on SiO2 nanoparticles genotoxicity.
Response:
We revised and added references 32-35 for discussion.
Minor remarks:
In SiO2, 2 should be in subscript.
Response:
We revised subscript 2 in SiO2 through the manuscript.
I would suggest using h/min instead of hours/minutes.
Response:
We revised h/min instead of hours/minutes in the manuscript.
Page 4, line 188 – change to mammalian cell line
Response:
We revised mammalian cell line on p4, para2.
Put Salmonella Typhimurium in italics.
Response:
We put all Salmonella Typhimurium in italics.
Round 2
Reviewer 1 Report
Just a note: If the Particle dimensions are obtained by TEM images, the values are the average of different measurement and must be reported with the error.
Author Response
Just a note: If the Particle dimensions are obtained by TEM images, the values are the average of different measurement and must be reported with the error.
Response: We are terribly sorry about the incomplete information regarding the particle sizes of these nanomaterials. Those nanomaterials were obtained from the Industrial Technology Research Institute (ITRI), and they only reported the average size of those materials. Given the research group of the materials was dissolved a few years back, we are very sorry that we don’t have the values of the standard error of the mean.
Reviewer 2 Report
The manuscript became somewhat more understandable, but the authors did not answer a number of questions.
Unanswered questions:
1. How were the nanomaterials obtained? How stable are they over time (in the form of a powder, in an aqueous solution)?
2. Did I understand correctly that the study was carried out in absolute darkness?
3. TEOS - not decrypted. "aTOES: tetraethoxysilane" is not used at all in the manuscript. Or is it just a change of letters in places? If yes, please correct!
4. The name of table 5 should be changed to display S9.
P.S. The authors answered some questions, but did not add any explanations to the text. I ask the authors to do this.
Author Response
The manuscript became somewhat more understandable, but the authors did not answer a number of questions.
Unanswered questions:
1. How were the nanomaterials obtained? How stable are they over time (in the form of a powder, in an aqueous solution)?
Response: The nanomaterials were obtained from the Industrial Technology Research Institute (ITRI). ITRI also provided us with the physical and chemical characteristic information of those materials, such as their TEM data. The nanomaterials were sent to us in their powder form, which was stable and kept away from light at all times. However, to reduce any instability that may occur during our experiments, we only weighted out power to make the solution enough for one experiment and used them immediately.
- Did I understand correctly that the study was carried out in absolute darkness?
Response: The nanomaterials were sent to us by ITRI in their powder form, which was stable and kept away from light at all times. However, to reduce any instability that may occur during our experiments, we only weighted out power to make the solution enough for one experiment and used them immediately.
- TEOS - not decrypted. "aTOES: tetraethoxysilane" is not used at all in the manuscript. Or is it just a change of letters in places? If yes, please correct!
Response: The silica nanoparticles were obtained by hydrolysis of tetraethoxysilane (TEOS) in an ethanol medium. We. Added the sentence in 2.1
The name of table 5 should be changed to display S9.
Response: We revised the name of table 5.
P.S. The authors answered some questions, but did not add any explanations to the text. I ask the authors to do this.
Response: We have added these explanations to the text in red word.